# Assessing Efficiency in Artificial Neural Networks

**Nicholas J. Schaub** [1,2,*,†] **and Nathan Hotaling** [1,2,†]

1  National Center for Advancing Translational Science, National Institutes of Health, Rockville, MD 20850, USA; nathan.hotaling@nih.gov
2  Axle Research and Technology, Rockville, MD 20850, USA
*  Correspondence: nick.schaub@nih.gov
†  These authors contributed equally to this work.

**Abstract:** The purpose of this work was to develop an assessment technique and subsequent metrics that help in developing an understanding of the balance between network size and task performance in simple model networks. Here, exhaustive tests on simple model neural networks and datasets are used to validate both the assessment approach and the metrics derived from it. The concept of neural layer state space is introduced as a simple mechanism for understanding layer utilization, where a state is the on/off activation state of all neurons in a layer for an input. Neural efficiency is computed from state space to measure neural layer utilization, and a second metric called the artificial intelligence quotient (aIQ) was created to balance neural network performance and neural efficiency. To study aIQ and neural efficiency, two simple neural networks were trained on MNIST: a fully connected network (LeNet-300-100) and a convolutional neural network (LeNet-5). The LeNet-5 network with the highest aIQ was 2.32% less accurate but contained 30,912 times fewer parameters than the network with the highest accuracy. Both batch normalization and dropout layers were found to increase neural efficiency. Finally, networks with a high aIQ are shown to be resistant to memorization and overtraining as well as capable of learning proper digit classification with an accuracy of 92.51%, even when 75% of the class labels are randomized. These results demonstrate the utility of aIQ and neural efficiency as metrics for determining the performance and size of a small network using exemplar data.

**Keywords:** neural networks; intelligence; neural efficiency; entropy; metrics





## 1. Introduction

The current trend in the artificial intelligence (AI) and machine learning (ML) field is to build larger artificial neural networks (ANNs) to increase task performance, with recent methods enabling training on models with trillions of parameters [1–3]. This trend stands in contrast to the traditional theory of over-parameterization in models leading to overfitting and worse performance. Additionally, it is now well established that within large, over-parameterized neural networks, low-dimensional manifolds exist that provide improved separability between classes as data pass deeper through the network [4]. It is these concepts that led to the experiments and metrics developed in this paper.

Successfully training large ANNs presents many challenges. Some of these challenges have found solutions in new types of layers or architectural designs. For example, some networks are too large for the task they are being trained to perform, resulting in poor task performance relative to a smaller network trained to perform the same task. This phenomenon is observed by increased performance on a task as the network depth increases to a point after which performance decreases. One solution to the costly search for the "optimal" network depth is skip or residual connections, which may require depth rescaling using another neuron layer [5]. Another major issue is overtraining/memorization, a common problem in nearly all modern neural networks. Overtraining causes ANNs to memorize inputs rather than learn a set of rules that generalize to new data, and it has

been shown that modern networks are highly capable of memorizing randomized image labels [6]. To combat overtraining and memorization, a variety of techniques and layers are used, including data augmentation [7], dropout layers [8], and even the creation of new neural networks in the case of adversarial neural network training [9]. Thus, the drive to create larger ANNs is often compounded by further increases in ANN size by using architecture components designed to resolve the problems of large networks, which make the deployment of these networks at scale a challenge [10].

The current state of ANN research has an analogous mindset to the human intelligence research prior to the formulation of the neural efficiency hypothesis in the 1990s [11]. The general opinion in neuroscience research was that an individual with high intelligence was more capable of performing tasks with high performance because their brain would be more capable of recruiting large numbers of neurons. This changed with publications that showed fewer neurons fire for a given task in higher performing individuals Haier et al. [12,13]. One publication showed that individuals with high intelligence had higher scores on Tetris but had lower brain metabolism while playing the game [11]. This finding led to the formulation of the neural efficiency hypothesis, which states that a key factor to intelligence is the capacity of the brain to perform a task by using the smallest amount of neural activity [14]. In the above context of human intelligence and neural efficiency, the current drive to increase neural network task accuracy by increasing the size and complexity of a network may be interpreted as the development of less intelligent neural networks. This is supported anecdotally in the literature by the tendency of most neural networks to memorize images with randomized labels during training [6].

Inspired by the discovery of the neural efficiency hypothesis from human intelligence research, this work develops an assessment technique called neural layer state space, which can be used to understand neural efficiency in simple neural networks. The ultimate goal of the neural state space assessment technique is to develop a method to measure over-parameterization within individual layers of simple neural networks. In addition to this, a metric that balances model performance and efficiency was developed and called the artificial intelligence quotient (aIQ), whose value is high when a small number of neurons generalize well to make accurate predictions on test data. This is a unique metric for neural networks because there are a large number of metrics for assessing neural network task performance [15,16] and a variety of metrics for assessing computational efficiency (wall time, model size, flops [17], algorithmic efficiency [18], etc.), but there are few to no metrics that balance task performance and efficiency.

To develop a mechanistic understanding of neural efficiency in small neural networks, two classic small neural networks are used: LeNet-300 and LeNet-5 [19]. The LeNet-300-100 is a fully connected network that, because the network only contains two hidden layers, permits an easy visualization of trends in model accuracy, aIQ, and the efficiency of each layer. LeNet-5 is a small convolutional neural network originally designed to work on MNIST. MNIST is used here since it is a well known and characterized dataset.

## 2. Approach

### 2.1. State Space

Prior attempts to increase the efficiency of a neural network were based on the removal of weights (pruning) based on weight magnitude or gradients of weights during backpropagation [20,21] or an analysis of firing frequency [22]. In this manuscript, we introduce the concept of neuron layer state space. The state space of a layer is the collection of neuron layer states, and a single state is the collective output of a neural layer given a single set of inputs. Since the output values of all neurons for a given set of inputs are generally passed to the subsequent layer, it was hypothesized that analyzing how neurons fire as a collective in a layer, rather than analyzing individual neurons, would be beneficial.

If the output of a neuron layer defines a single state, then the state space is the set of all observed states in a layer when inputs from the train or test data pass through the network. When one image is passed through a convolutional neural network, convolutional layers

will generate multiple neuron layer states per image (micro states) that make up a spatial activation map (a macro state). In contrast, dense layers will generate only one neuron layer state per image (the micro and macro state are the same). In this manuscript, to assess a neuron's state, its outputs are quantized as either firing (output is greater than zero) or non-firing (output is less than or equal to 0). However, even with quantization, the state space could still be unmanageably large since the number of possible states in a layer after quantization will be $2^{N_l}$, where $N_l$ is the number of neurons in a layer. In reality, significantly fewer states are actually generated in a layer, so bins for a layer state are only created when observed.

*2.2. Neural Efficiency*

Neural efficiency is defined as the utilization of state space, and can be measured by entropic efficiency. If all possible states are recorded for data fed into the network, then the probability, $p$, of a state occurring can be used to calculate Shannon's entropy, $E_l$, of network layer $l$:

$$E_l = - \sum_{i=0}^{i=N_l} p_i * log_2(p_i) \tag{1}$$

Intuitively, $E_l$ is an estimation of the minimum number of neurons required to encode the information exported by the neural layer if the output information could be perfectly encoded. The maximum theoretical entropy of the layer will occur when all states occur the same number of times, and the entropy value will be equal to the number of neurons in the layer, $N_l$. Neural efficiency, $\eta_l$, can then be defined as the entropy of the observed states ($E_l$) relative to the maximum entropy ($N_l$):

$$\eta_l = \frac{E_l}{N_l} \tag{2}$$

Thus, neural efficiency, $\eta_l$, is defined as state space efficiency using Shannon's entropy with a range of 0–1. Neural efficiency values close to zero are likely to have more neurons than needed to process the information in the layer, while neuron layers with neural efficiency close to one are making maximum usage of the available state space. Alternatively, very high neural efficiency could also mean too few neurons are in the layer.

*2.3. Artificial Intelligence Quotient*

Neural efficiency is a characteristic of intelligence, but so is task performance. Therefore, an intelligent algorithm should perform a task with high accuracy and efficiency. Using $\eta_l$ as layer efficiency, the neural network efficiency, $\eta_N$, can be calculated as the geometric mean of all layer efficiencies in a network containing $L$ number of layers:

$$\eta_N = \left( \prod_{l=1}^{L} \eta_l \right)^{\frac{1}{L}} \tag{3}$$

The geometric mean was chosen to average efficiencies since the geometric mean is robust to outliers. The artificial intelligence quotient (aIQ) can thus be defined as

$$aIQ = \left( P^\beta * \eta_N \right)^{\frac{1}{\beta+1}} \tag{4}$$

where $P$ is the performance metric, and $\beta$ is a tuning parameter to give more or less weight to performance at the cost of $\eta_N$.

**3. Experiments**

*3.1. Exhaustive LeNet Training*

To evaluate neural efficiency and aIQ, two types of neural networks were trained on the MNIST digits dataset [19]. The first network (LeNet-300-100) consists of two densely

connected layers followed by a classification layer. The second network (LeNet-5) consisted of two convolutional layers, each followed by a max pooling layer ($2 \times 2$ pooling, stride 2), and a densely connected layer followed by a classification layer. All layers used exponential linear unit (ELU) activation [23] and L2 weight regularization (0.0005). Standard stochastic gradient descent with Nesterov updates was used with a static learning rate of 0.001 and a momentum of 0.9. Training was stopped when the training accuracy did not increase within five epochs of a maximum value.

For each neural network architecture, the number of neurons in every layer varied from 2 to 1024 by powers of 2. All combinations of layer sizes were trained, with 11 replicates using different random seeds to determine variability resulting from different initializations. A total of 11,000 models were trained for the LeNet-5 architecture, and 1100 were trained for the LeNet-300-100 architecture. Models were constructed and trained using Tensorflow 2.7, and networks were trained in parallel on two gpu servers with 8 NVidia Quadro RTX 8000s each. Once models were trained, aIQ was calculated with $\beta = 2$ to give a nominal preference for networks with higher accuracy.

### 3.2. Dropout and Batch Normalization

In the context of neural layer efficiency, batch normalization was hypothesized to be a method to improve efficiency, while dropout is a method to decrease efficiency. The rationale for batch normalization improving neural efficiency is that neuron activation is driven toward the center of the distribution of neuron outputs. As the firing frequency of each neuron approaches 50%, the entropy is more likely to obtain the maximum value. In contrast, dropout was hypothesized to decrease the available state space during training by dropping the outputs of neurons, effectively decreasing the maximum entropy value. To test the effect of dropout and batch normalization, neural networks were trained with the same number of neurons, as described in the Exhaustive LeNet Training section, except a dropout layer ($p = 50\%$) or batch normalization layer was added after every hidden layer. For networks with batch normalization or dropout layers, only three replicates were trained. A total of 3000 networks were trained each for LeNet-5 networks with either dropout or batch normalization. A total of 300 networks were trained each for LeNet-300-100 networks with either dropout or batch normalization.

### 3.3. Memorization and Generalization Tests

If aIQ provides an assessment of capacity to learn general rules rather than memorize training inputs, it was hypothesized that network architectures with a high aIQ would perform well when trained on datasets with noisy labels. The reason for this is that, for networks with a low aIQ and with low $\eta_N$, training inputs are memorized because the state space is likely much larger than the space of observed states. This means that new data would be classified correctly or incorrectly based on how similar an image was to one of the memorized inputs. However, for networks with a high aIQ, there is insufficient bandwidth to create a special state for an input with a low prevalence label. To test this, network architectures were trained from scratch where 25%, 50%, 75%, or 100% of the training labels were randomized. Accuracy and neural efficiency was then measured for both the test and train datasets without randomized labels. To test the capacity of high aIQ networks to generalize to new data, trained networks were used to evaluate the EMNIST dataset [24], which contains 280,000 additional digit images in the same formats as the original 70,000 digit images contained in MNIST. Three thousand networks were trained for the LeNet-5 architecture for each of the different percentages of randomization (12,000 in total) and each type of layer modifier (batch normalization, dropout, or neither). Three hundred networks were trained for each combination of LeNet-300-100 architecture and label randomization.

## 4. Results

### 4.1. Accuracy, aIQ, and Neural Efficiency

Testing 81,004 model combinations was performed to understand trends in aIQ, neural efficiency, and accuracy. For the training data (not shown), accuracy increased monotonically with the number of neurons added in each layer, but the test data showed a slight decrease in accuracy as the number of neurons in Hidden Layer 1 ($N_1$) contained more than 128 neurons (Figure 1a). In contrast, aIQ ($\beta = 2$) values reached a local maximum when $N_1 = 8$ and $N_2 = 4$ (Figure 1b). The decrease in aIQ values are due to the trend in neural efficiency to decrease as the number of neurons in a layer increases (Figure 1c,d). It was generally observed that neural efficiency was inversely related to the number of neurons, and changing the number of neurons in one layer had an impact on the neural efficiency of other layers. For example, for networks where $N_1 = 8$, the neural efficiency of Layer 1, $\eta_1$, generally increased as $N_2$ increased, except there was a local minimum at $N_2 = 16$. Local minima can be observed in other areas where the number of neurons in either layer is held constant, and the changes in efficiency for the same layer are observed (e.g., when $N_2 = 8$, a local minima occurs when $N_1 = 32$). Similar trends were observed in the LeNet-5 models, where changes in the number of neurons in one layer affected the efficiency of other layers.

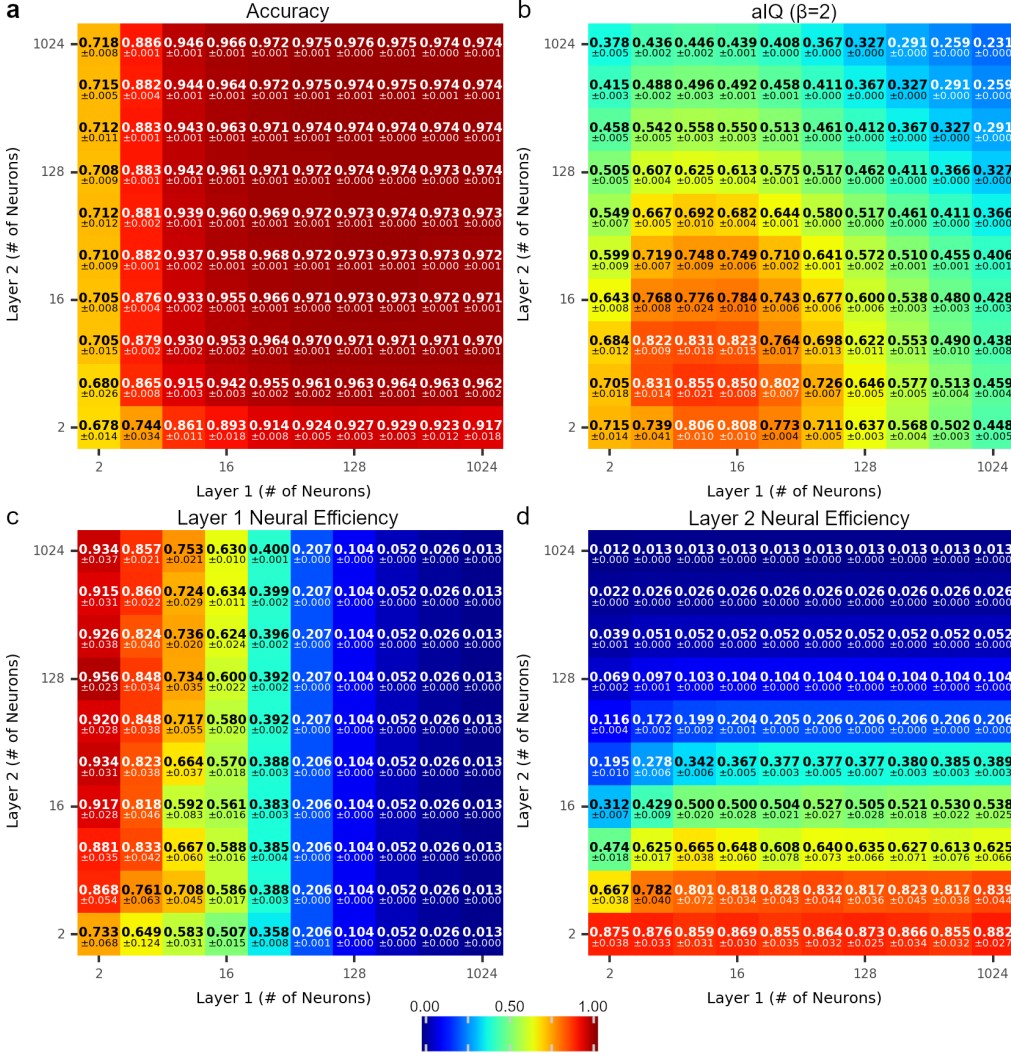

**Figure 1.** Trends in accuracy (**a**), aIQ (**b**), and neural efficiency for Hidden Layer 1 (**c**) and Hidden Layer 2 (**d**) in the LeNet-300-100 model. Data shown are for test data. All values range from 0 to 1, and values are the mean ±95% confidence interval of 11 replicates.

Additional experiments were performed to identify networks with the highest aIQ for each architecture. Analysis of the top three neural networks for accuracy or aIQ for both LeNet models are shown in Table 1. For the LeNet-300-100 models, the model with the highest test accuracy ($N_1 = 128$, $N_2 = 1024$) achieved an accuracy of 97.58% $\pm$ 0.04% with an aIQ of 32.7 $\pm$ 0.01 (values are mean $\pm$ 95% CI, n = 11). The highest aIQ model ($N_1 = 11$, $N_2 = 4$) had an accuracy of 92.91% $\pm$ 0.19% with an aIQ of 86.41 $\pm$ 0.71. Thus, the network with the highest aIQ was 4.76% less accurate but contained 3.60% of the parameters.

The differences between the networks with the highest accuracy and those with the highest aIQ were even more drastic for the LeNet-5 models. The network with the highest accuracy ($N_1 = N_2 = N_3 = 1024$) had an accuracy of 99.58% $\pm$ 0.02% and an aIQ of 24.28 $\pm$ 0.004 (values are mean $\pm$95% CI, n = 11). However, the network with the highest aIQ ($N_1 = 3$, $N_2 = 9$, $N_3 = 4$) had an accuracy of 96.84% $\pm$ 0.18% and an aIQ of 88.28 $\pm$ 0.46. The network with the highest aIQ had a lower accuracy by 2.32% but contained 0.0032% of the parameters.

**Table 1.** The top three network architectures by accuracy and aIQ.

| LeNet-300-100 ($N_l$) | | Test Accuracy (%) [†] | aIQ ($\beta = 2$) [†] | Parameters (fold decrease) |
|---|---|---|---|---|
| Layer 1 | Layer 2 | | | |
| 128 | 1024 | **97.58 $\pm$ 0.04** | 32.70 $\pm$ 0.01 | 242,826 (1×) |
| 256 | 1024 | 97.54 $\pm$ 0.06 | 29.12 $\pm$ 0.01 | 474,378 (0.5×) |
| 64 | 1024 | 97.53 $\pm$ 0.08 | 36.67 $\pm$ 0.02 | 127,050 (1.9×) |
| 11 | 4 | 92.91 $\pm$ 0.19 | **86.41 $\pm$ 0.71** | 8733 (27.8×) |
| 11 | 5 | 93.59 $\pm$ 0.29 | 85.93 $\pm$ 1.04 | 8755 (27.7×) |
| 7 | 4 | 90.76 $\pm$ 0.25 | 85.90 $\pm$ 1.00 | 5577 (43.5×) |

| LeNet-5 ($N_l$) | | | Test Accuracy (%) [†] | aIQ ($\beta = 2$) [†] | Parameters (fold decrease) |
|---|---|---|---|---|---|
| Layer 1 | Layer 2 | Layer 3 | | | |
| 1024 | 1024 | 1024 | **99.16 $\pm$ 0.02** | 24.28 $\pm$ 0.004 | 43,030,538 (1×) |
| 1024 | 128 | 512 | 99.15 $\pm$ 0.03 | 33.02 $\pm$ 0.008 | 4,357,770 (9.9×) |
| 1024 | 512 | 1024 | 99.14 $\pm$ 0.03 | 26.22 $\pm$ 0.005 | 21,534,218 (2.0×) |
| 3 | 9 | 4 | 96.84 $\pm$ 0.18 | **88.28 $\pm$ 0.46** | 1392 (30,912.9×) |
| 3 | 5 | 5 | 96.72 $\pm$ 0.16 | 88.18 $\pm$ 1.52 | 923 (46,620.3×) |
| 3 | 4 | 4 | 95.37 $\pm$ 0.19 | 88.02 $\pm$ 0.64 | 692 (62,182.9×) |

† Accuracy and aIQ values are mean $\pm$95% CI (n = 11). aIQ values are ×100. Data shown are for metrics calculated on the test dataset.

### 4.2. Batch Normalization and Dropout as Neural Efficiency Modifiers

#### 4.2.1. Batch Normalization

Batch normalization generally increased the accuracy and $\eta_N$ for both LeNet-300-100 and LeNet-5 networks, resulting in a rise in aIQ for most networks (Table 2). For LeNet-300-100 networks, 178 (59.26%) of the networks with batch normalization had a mean $\eta_N$ higher than the mean $\eta_N$ of corresponding networks trained without batch normalization. When only small networks are considered ($N_1 \leq 16$, $N_2 \leq 16$), 36 (74.13%) of the networks with batch normalization had a mean $\eta_N$ higher than the same network trained without batch normalization. In addition to higher network efficiency, neural networks trained with batch normalization also had higher accuracy on test data, where 233 (77.78%) of the networks with batch normalization achieved higher accuracies than the same networks without batch normalization. aIQ increased in 222 (74.07%) of the networks where all layers had 16 or fewer neurons.

For LeNet-5 networks, 2226 (74.2%) of all networks with batch normalization had a mean $\eta_N$ higher than their corresponding networks without batch normalization. However, the batch normalization generally resulted in lower accuracy, with only 978 (32.60%) of the networks achieving a higher accuracy than their corresponding networks without batch normalization. When considering only small networks where $N_l \leq 16$ for all layers, 178 (92.6%) of the networks with batch normalization had a higher accuracy. Furthermore,

2133 (71.10%) of all networks with batch normalization had a higher $\eta_N$ relative to their corresponding networks without batch normalization, and $\eta_N$ decreased to 123 (64.35%) for networks where all layers had less than 16 neurons. Overall, 2226 (74.20%) of all networks had a higher aIQ when trained with batch normalization.

These results confirm the hypothesis that batch normalization generally acts to improve $\eta_N$ in addition to improving classification accuracy (Table 2), making neural networks "more intelligent", as assessed by aIQ.

**Table 2.** The top network architectures by aIQ when batch normalization (BatchNorm), dropout (Dropout), or neither (None) are added to every layer.

| LeNet-300-100 ($N_l$) | | | | | |
|---|---|---|---|---|---|
| Layer 1 | Layer 2 | Modifier | Test Accuracy (%) [†] | aIQ ($\beta = 2$) [†] | $\eta_N$ (%) |
| 11 | 4 | **None** | 92.91 ± 0.19 | **86.41 ± 0.71** | **74.77 ± 1.72** |
| | | BatchNorm | **93.50 ± 0.36** | 82.10 ± 1.44 | 63.33 ± 3.07 |
| | | Dropout | 78.41 ± 3.82 | 74.36 ± 4.06 | 66.89 ± 4.47 |
| 10 | 6 | None | 93.52 ± 0.14 | 83.16 ± 2.52 | 66.20 ± 5.75 |
| | | **BatchNorm** | **93.65 ± 0.44** | **87.76 ± 0.44** | **77.07 ± 1.16** |
| | | Dropout | 83.63 ± 0.63 | 75.62 ± 0.93 | 61.83 ± 1.49 |
| 7 | 5 | None | 91.53 ± 0.31 | 83.88 ± 1.75 | 70.65 ± 4.21 |
| | | BatchNorm | **91.91 ± 0.25** | **86.42 ± 1.20** | **76.43 ± 2.76** |
| | | **Dropout** | 80.63 ± 2.96 | 78.31 ± 2.44 | 73.88 ± 1.47 |

| LeNet-5 ($N_l$) | | | | | | |
|---|---|---|---|---|---|---|
| Layer 1 | Layer 2 | Layer 3 | Modifier | Accuracy (%) [†] | aIQ ($\beta = 2$) [†] | $\eta_N$ (%) |
| 3 | 9 | 4 | **None** | 96.84 ± 0.18 | **88.28 ± 0.46** | 73.38 ± 1.22 |
| | | | BatchNorm | **96.95 ± 0.58** | 82.44 ± 3.08 | 59.73 ± 6.47 |
| | | | Dropout | 79.72 ± 4.96 | 77.82 ± 2.91 | **74.23 ± 0.99** |
| 2 | 8 | 8 | None | 97.83 ± 0.10 | 85.58 ± 2.28 | 65.81 ± 5.03 |
| | | | **BatchNorm** | **98.05 ± 0.20** | **88.12 ± 1.68** | 71.21 ± 4.30 |
| | | | Dropout | 88.06 ± 2.00 | 82.75 ± 0.38 | **73.16 ± 4.21** |
| 3 | 4 | 7 | None | 97.17 ± 0.22 | 83.71 ± 2.65 | 62.55 ± 5.74 |
| | | | BatchNorm | **97.27 ± 0.34** | 84.91 ± 2.16 | 64.76 ± 4.45 |
| | | | **Dropout** | 90.15 ± 1.66 | **85.45 ± 1.47** | **76.82 ± 3.32** |

[†] Accuracy, aIQ, and efficiency values are mean ±95% CI (n = 11 for None, n = 3 otherwise). aIQ values are ×100. Data shown are for metrics calculated on the test dataset.

### 4.2.2. Dropout

Dropout had different effects on the LeNet-300-100 and LeNet-5 architectures. Dropout generally decreased $\eta_N$ in LeNet-300-100 networks and increased $\eta_N$ in the LeNet-5 networks, but decreased aIQ in nearly all networks due to a drop in accuracy for nearly all networks (Table 2). For LeNet-300-100 networks, none of the networks trained with dropout had an accuracy higher than their corresponding networks trained without dropout, and only 18 (5.59%) of the networks had a higher efficiency. Accordingly, no networks with dropout had a higher aIQ relative to their corresponding networks without dropout.

None of the networks with dropout had an accuracy higher than their corresponding networks without dropout, and 1818 (60.60%) of the dropout networks had a higher $\eta_N$. When considering neural networks with 16 or fewer neurons in each layer, 191 (99.33%) of the networks with dropout had a higher $\eta_N$ than their corresponding networks without dropout. However, only 468 (15.60%) of the dropout networks had a higher aIQ than their corresponding networks without dropout, meaning the increase in efficiency was not sufficient to offset the decrease in accuracy in the majority of networks.

The general conclusion from these results is that dropout generally decreases accuracy, leading to a drop in aIQ. The reason why accuracy dropped in all networks may be due to a variety of factors, including the dropout rate being too high ($p = 50\%$) or dropout being placed in every layer of the network. The explanation for why dropout appeared

to decrease $\eta_N$ in LeNet-300-100 networks but increased $\eta_N$ in LeNet-5 networks may be due to LeNet-300-100 only having dense layers and LeNet-5 containing convolutional layers. It is known that dropout can have a nominal or detrimental impact on network performance because dropout layers add noise to the network [25]. As a result, the increased $\eta_N$ may be due to increased noise in the convolutional layers, leading to higher entropy. A better assessment of the effect of dropout for convolutional layers may be to use dropout layers with different dropout rates or use dropout layer types constructed specifically for convolutional layers, such as dropblock or spatial dropout (a survey of different dropout types has been performed [26]).

### 4.3. Memorization and Generalization

To test the resistance of networks to overfitting/memorization, a percentage of image labels were randomized before training, as previously described by Zhang et al. [6]. Over-fitting occurs when incorrect, random labels are learned and is analogous to memorizing the label for an image. Table 3 shows the results for the neural networks with the highest aIQ and the highest accuracy from the batch normalization tests for both the LeNet-300-100 and LeNet-5 networks. When none of the labels are randomized (0%), the accuracy of the largest network is higher than the network with the best aIQ for both LeNet-300-100 and LeNet-5. When 25–75% of the labels were randomized, the network with the highest aIQ had the best accuracy. Even when 75% of the labels were randomly assigned, the networks with a high aIQ were able to achieve accuracies of 84.68% and 92.51% for the LeNet-300-100 and LeNet-5 networks, respectively. These numbers are considerably better than the neural networks with the highest accuracies in the batch normalization tests, which had test accuracies of 36.99% and 44.24% when trained on data with 75% of the labels randomized. Both types of networks had poor performance regardless of original aIQ or accuracy values when all labels were randomly assigned (100%). This result demonstrates that networks with a high aIQ have the property of being resistant to overtraining and memorization and that they can learn the correct classification weights even when a majority of the input labels are incorrect.

To test the capacity of networks to generalize results to a larger, more diverse dataset, the networks with the highest aIQ or the highest accuracy were trained on the MNIST dataset, and classification accuracy was then measured on the EMNIST dataset. The EMNIST digits dataset has a similar format to MNIST, but it contains 280,000 more samples [24]. The general trend was that the network with the highest accuracy showed a superior performance on EMNIST in comparison to MNIST, with significant differences between the LeNet-300-100 and LeNet-5 networks, Table 4). The LeNet-300-100 network with the highest accuracy on MNIST (98.03%) showed a much greater performance than that with the highest accuracy on EMNIST (76.21%), while the network with the highest aIQ showed a much larger decrease in accuracy from MNIST to EMNIST (from 93.65% to 58.43%, respectively). In contrast, the differences between the networks with the highest accuracy and the highest aIQ were less drastic among the LeNet-5 networks. The EMNIST accuracy was 90.56% in the network with the highest accuracy and was 89.36% in the network with the highest aIQ. These experiments were repeated after training the same networks on MNIST with 75% of the labels randomized, and as expected the networks with the highest accuracy showed a significant decrease in performance on EMNIST, while the networks with the highest aIQ showed a considerably improved performance on both the LeNet-300-100 and LeNet-5 networks. This data demonstrate that convolutional neural networks with high aIQ do not considerably underperform in comparison to much larger networks when performance is assessed on a much larger, diverse dataset, but dense networks with a high aIQ may not generalize as well.

**Table 3.** Memorization tests on the networks with the highest aIQ or accuracy.

| LeNet-300-100 ($N_l$) | | Accuracy (%) | | | | |
|---|---|---|---|---|---|---|
| Layer 1 | Layer 2 | 0% Rand | 25% Rand | 50% Rand | 75% Rand | 100% Rand |
| 1024 | 1024 | 98.03 | 85.91 | 63.43 | 36.99 | 9.48 |
| 10 | 6 | 93.65 | 91.56 | 89.59 | 84.68 | 10.89 |
| LeNet-5 ($N_l$) | | | | | | |
| Layer 1 | Layer 2 | Layer 3 | 0% Rand | 25% Rand | 50% Rand | 75% Rand | 100% Rand |
| 1024 | 1024 | 1024 | 99.11 | 93.99 | 76.07 | 44.24 | 10.34 |
| 2 | 8 | 8 | 98.05 | 96.73 | 95.69 | 92.51 | 7.49 |

The % Rand indicates the percentage of labels that were randomized prior to training.

**Table 4.** Generalization tests on the networks with the highest aIQ or accuracy.

| LeNet-300-100 ($N_l$) | | MNIST Accuracy (%) | | EMNIST Accuracy (%) | |
|---|---|---|---|---|---|
| Layer 1 | Layer 2 | 0% Rand | 75% Rand | 0% Rand | 75% Rand |
| 1024 | 1024 | 98.03 | 36.99 | 76.21 | 24.93 |
| 10 | 6 | 93.65 | 84.68 | 58.43 | 55.46 |
| LeNet-5 ($N_l$) | | | | | |
| Layer 1 | Layer 2 | Layer 3 | 0% Rand | 75% Rand | 0% Rand | 75% Rand |
| 1024 | 1024 | 1024 | 99.11 | 44.24 | 90.56 | 34.16 |
| 2 | 8 | 8 | 98.05 | 92.51 | 89.36 | 74.67 |

## 5. Discussion

The major contribution of this work is the establishment of an assessment technique called neuron layer state space and the capacity to use state space as a means of evaluating neuron layer utilization using the metrics of neural efficiency and aIQ. Quantizing neurons into on/off positions that encode information collectively as a neural layer state provides a different perspective on how data are processed in the network. One prevailing thought is that neurons are discrete units that encode individual features, but the work here suggests that the information an individual neuron encodes may have value in the context of the other neurons with which it fires. One advantage of conceptualizing the flow of information between layers using state space is the number of tools that become available for network analysis. In this work, a rudimentary metric was created to understand layer utilization, but many other methods of analyzing state space could be used, such as the relative entropy of a single layer or the mutual information between two layers. Further investigation of state space may help to further compress the size of the network without a significant decrease in accuracy, and may even permit the training of the number of neurons in a layer. While neural architecture search has become a topic of interest in order to search for an ideal network computational cell, few if any of these methods include parameters to learn layer sizes.

One benefit of understanding neural networks in terms of state space is that guidelines on how many neurons to place in a layer can be established knowing only superficial information about the training data. The current thought is that a higher number of neurons leads to higher accuracy, but this does not appear to provide an improved generalization for the small convolutional models tested here (see Table 4). Using the concept of state space, the number of states in a dense layer cannot exceed the number of input images. If there are X training examples, then $ceil(log_2 X)$ neurons are sufficient to memorize every training image. As an example, the ImageNet dataset has 14 million images (leading to $\sim 2^{24}$ states), which is considerably smaller than the available statespace of AlexNet (4096 densely connected neurons), which is $\sim 6 \times 10^{1225}$ times larger than the number of available training examples in ImageNet [27]. Due to random initialization, it is doubtful

that a dense layer would memorize every training image if it contains exactly the number of neurons required to memorize all inputs. However, assuming the network is generating general rules for classification, the entire bandwidth of the channel should never need to be used. An analogous guideline could be applied to convolutional layers, where all combinations of pixel intensities of an 8-bit grayscale image in a $3 \times 3$ grid could be perfectly represented by 72 neurons with $3 \times 3$ convolutional kernels ($256^9 = 2^{72}$), so the first convolutional layer of a network should never contain more than 72 neurons when analyzing 8-bit grayscale images. Thus, simple upper limits on the number of neurons in different layers could be inferred from the implications of state space, where the upper limits may be considerably smaller than the number of neurons that are currently observed in modern networks.

Using the guidelines laid out above, it is reasonable to say that most networks that are created contain many more neurons than needed, explaining why most neural networks are prone to memorization and attack vectors. The authors of Morcos et al. [28] demonstrated the existence of single directions in neural networks, and although they found that models trained on randomized labels did not generalize, they did find that single directions were significantly more important in models with randomized labels for training data accuracy [28]. In the present work, we constrained the state space by reducing the number of neurons in a layer. If memorization is the construction of unique states within state space, then constraining the number of available states in a layer should reduce or even prevent memorization by limiting the number of potential single directions. For attack vectors, adversarial networks can be trained to add imperceptible amounts of noise to an image to cause the neural network to misclassify the image [29]. It is plausible that these attack vectors take advantage of noise in an over-parameterized network, a problem that a smaller neural network may not face. Thus, the use of state space and neural efficiency may help to make networks more resistant to such attacks.

Related to the size complexity of current neural networks is the concept of Occam's razor in machine learning. The general concept behind Occam's razor is that the simplest model should be preferred, including in artificial intelligence [30]. Geometric analysis of the parameter space of neural networks reveals low-dimensional manifolds, suggesting that, despite the vast number of parameters in most modern neural networks, they have relatively simple parameter spaces [31]. Thus, the work presented here evaluating efficiency could be interpreted as a metric that assesses model simplicity.

There are a few deficits in the current approach that should be resolved in future work. The first is the memory utilization and computational complexity of capturing state space. Only small networks were analyzed in this work due to the potential size of state space, which may be unmanageable in a large neural network assessing a large dataset such as ImageNet.

The second problem is the issue of class imbalances. If there are class imbalances in either the training or test data, it would be expected that the efficiency metrics would be skewed. Class imbalances in the training data might cause more neurons in the network to be dedicated to classification of the most common class, while imbalances in the test data might overrepresent states that occur. Another issue is the underlying assumption that maximum efficiency is achieved when all states occur at the same frequency. It might be expected that some states occur far more frequently than others so that an ideal distribution of states might look more like an exponential distribution. This was superficially confirmed by looking at the distribution of states in the LeNet-300-100 networks, by analyzing networks with $N_1 = 8$ around the local minimum when $N_2 = 16$ (see Figure 1). Therefore, some other metric of calculating efficiency that accounts for an ideal distribution of states might be a better measure of efficiency.

The third deficit is the issue of implementation. Recording and processing data collected in state space is expensive. For some networks, the number of neurons in a convolutional layer can be 128 or more, meaning that at least $2^{128}$ layer states are possible and that the states for every location in an image should be tracked to calculate the entropy.

The current implementation of calculating entropy is not practical for larger networks that contain many more layers and layers with larger numbers of neurons (such as AlexNet with 4096 neurons in the dense layers) [27]. One potential solution to this might be to create a method of approximating the entropy by, rather than recording every observed state, collecting sufficient information on a layer-by-layer basis to capture the shape of the distribution.

Finally, one topic not investigated here is how data augmentation impacts efficiency and aIQ. Data augmentation is generally used to help improve accuracy and generalization, likely because it helps to mitigate memorization. Networks with high aIQ were small and were fairly resistant to memorization (see Table 3), so it might be expected that certain types of augmentation (i.e., random cropping) might not improve performance when training a network with a high aIQ, but other types of augmentation might help (i.e., image flipping).

## 6. Conclusions

This work introduces an assessment technique called neuron layer state space, and demonstrates how an analysis of state space, using metrics such as neural efficiency and state space, is useful for assessing the trade-off between ANN accuracy and performance in simple architectures. Dense convolutional models with a high aIQ were shown to have desirable properties, such as resistance to overtraining and a general performance on EMNIST that is comparable to much larger networks. Future work with state space should establish metrics of layer efficiency as well as methods of computing efficiency and evaluating larger networks that are superior to the two models presented in this paper. State space may provide insight into sizing neural network layers to vastly decrease the size of existing neural networks.

**Author Contributions:** Conceptualization, N.J.S.; methodology, N.J.S.; software, N.J.S.; formal analysis, N.J.S.; investigation, N.J.S.; writing—original draft preparation, N.J.S.; writing—review and editing, N.H.; visualization, N.J.S.; supervision, N.H.; project administration, N.H. All authors have read and agreed to the published version of the manuscript.

**Funding:** This research received no external funding.

**Institutional Review Board Statement:** Not applicable.

**Informed Consent Statement:** Not applicable.

**Data Availability Statement:** The MNIST dataset was obtained through the torchvision datasets. The state capture, efficiency, and aIQ calculations are available as a public Python package called TensorState. The code is available at: https://github.com/Nicholas-Schaub/TensorState.

**Conflicts of Interest:** The authors declare no conflict of interest.

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
