# Peer review of "Assessing Efficiency in Artificial Neural Networks"

_applsci, doi:10.3390/app131810286_

Round 1
Reviewer 1 Report
The authors propose an assessment metric to measure the balance between network size and task performance. The paper seems interesting and the introduction is well written. Experimental assessment is convincing and well argued. However, there are several points that should be improved:
1) introduction: "For example, some networks are too large for the task they are being trained to perform, resulting in poor task performance relative to a smaller network trained to perform the same task." These sentence does not sound. Why I should prefer a larger network respect to a smaller one if the smaller network has greater performance?
2) section 2.1: "the state space is the frequency that each state of a layer occurs as all images in the train or test data pass through the network." I don't understand this sentence: why do the authors define the space as the frequency of each state? Intuitively, the space is the set of all the possible states, not the frequency of each of them. In the subsequent discussion about the dimension of the size space, it seems to agree with this intuitive definition, and not with the one given by the authors, where the term "frequency" is used. Is this an error in the definition or the authors give a different meaning to the term "space"? Please better specify this point, and if the term "space" is actually appropriate in this context.
3) section 2.1: "When one image is passed through a convolutional neural network, convolutional layers will generate multiple states per image. In contrast, dense layers will generate only one state per image." I don't agree with this point. A convolutional layer can be viewed as a dense layer with shared weights and broken connection, therefore both dense and convolutional layers should, in my opinion, generate the same number of states (in my opinion, just one) per input.
4) Section 2.2: the equation of the Entropy of a network layer should be better specified (extremes of the sum etc.).
5) section 2.3: the authors use the geometric mean of the layers efficiency to define the network efficiency. Why was the geometric mean adopted and not a simple mean? Please specify the reason of this choice.
6) the authors define the the artificial intelligence quotient as the geometric mean (again) of the entropy with the performance (that, I presume, it is assumed between 0 and 1). I think that adopting the term "artificial intelligence quotient" to define this measurement could be misleading, especially if compared with the classical definition of intelligence quotient applied usually, that it is not said that it is a direct consequence of the neural efficiency, since it is usually measured regardless the neural efficiency, considering other factors (such as age, intelligence tests,...). Therefore, defining the proposed measurement as "artificial intelligence quotient" could not be appropriate.
7) title: "Assessing Intelligence in Artificial Neural Networks" for the reason given above, I think the title is too daring...
Author Response
We would like to thank the reviewer for the thoughtful and detailed reading of our manuscript. Below we have responded to specific comments, with the original comment in italics and our response in bold.
1) introduction: "For example, some networks are too large for the task they are being trained to perform, resulting in poor task performance relative to a smaller network trained to perform the same task." These sentence does not sound. Why I should prefer a larger network respect to a smaller one if the smaller network has greater performance?
The next sentence after what the reviewer quotes in our paper answers the question the reviewer is asking. Large networks that do not have skip connections tend to perform worse that small networks (see reference 5). The reason to prefer large networks is that they tend to have more inference power, but if they are too deep and without skip connections, the performance can be bad.
2) section 2.1: "the state space is the frequency that each state of a layer occurs as all images in the train or test data pass through the network." I don't understand this sentence: why do the authors define the space as the frequency of each state? Intuitively, the space is the set of all the possible states, not the frequency of each of them. In the subsequent discussion about the dimension of the size space, it seems to agree with this intuitive definition, and not with the one given by the authors, where the term "frequency" is used. Is this an error in the definition or the authors give a different meaning to the term "space"? Please better specify this point, and if the term "space" is actually appropriate in this context.
This is an excellent criticism. As the reviewer correctly points out, state space should be the available states observed in the network, not the frequency of those states. We revised the manuscript to reflect this key point. More specifically, we have updated the text quoted by the reviewer so that it now reads as follows:
"...the state space is the set of all observed states in a layer when inputs from the train or test data pass through the network."
3) section 2.1: "When one image is passed through a convolutional neural network, convolutional layers will generate multiple states per image. In contrast, dense layers will generate only one state per image." I don't agree with this point. A convolutional layer can be viewed as a dense layer with shared weights and broken connection, therefore both dense and convolutional layers should, in my opinion, generate the same number of states (in my opinion, just one) per input.
If we understand the reviewer correctly, the reviewer is correct that each location in an image convolved with a filter has one input per region. However, for an image there are multiple locations that the filter is evaluated. For example, if an image is of size 7x7 and convolved with at 3x3 filter, if no padding is added to the original image then the output of the operation is a 5x5 image. This means there are 25 different states, one for each location, for a single input image. In contrast, a dense layer aggregates all input locations and produces one state.
To better convey this concept, we have rewritten the section so that it now reads:
"When one image is passed through a convolutional neural network, convolutional layers will generate a many states, one state for each location of the convolution. In contrast, dense layers will generate only one state per image."
4) Section 2.2: the equation of the Entropy of a network layer should be better specified (extremes of the sum etc.).
We are not certain what the reviewer is requesting here, and we made an attempt to respond to this based on our interpretation of the comment. We believe the reviewer is asking that we add summation limits, which we have included in this revision. If this is not what the reviewer is requesting, we are happy to make any subsequent modifications to improve the presentation of the equation.
5) section 2.3: the authors use the geometric mean of the layers efficiency to define the network efficiency. Why was the geometric mean adopted and not a simple mean? Please specify the reason of this choice.
The following sentence was added below the efficiency equation to justify the use of geometric mean:
The geometric mean was chosen to average efficiencies since the geometric mean is robust to outliers.
6) the authors define the the artificial intelligence quotient as the geometric mean (again) of the entropy with the performance (that, I presume, it is assumed between 0 and 1). I think that adopting the term "artificial intelligence quotient" to define this measurement could be misleading, especially if compared with the classical definition of intelligence quotient applied usually, that it is not said that it is a direct consequence of the neural efficiency, since it is usually measured regardless the neural efficiency, considering other factors (such as age, intelligence tests,...). Therefore, defining the proposed measurement as "artificial intelligence quotient" could not be appropriate.
Thank you for the comment. This is something we wrestled with as well, and the reason we decided to stay with aIQ is that the metric approximates features of human intelligence as described in the introduction. In a previous version we had additional background and discussion about the nuance of what aIQ is really is a reflection of, and that should be more along the lines of crystalized intelligence or task specific intelligence. However, we found that discussion to be fairly meandering and distracting. We maintain artificial intelligence quotient is still an appropriate name, because high values indicate a balance of task performance and efficiency, key features of intelligence in the human brain.
7) title: "Assessing Intelligence in Artificial Neural Networks" for the reason given above, I think the title is too daring...
As with the previous comment from the reviewer, we also spent a lot of time debating the title. Unlike the previous comment with the reviewer, we agree that this needs to be changed because it is "too daring". Using intelligence generically as we do in the title is misleading, but the use of artificial intelligence quotient is still acceptable because of the explanation and context we provide in the manuscript.
Thus, we have changed the title of the paper to "Assessing Efficiency in Artificial Neural Networks"
Reviewer 2 Report
Equations numbering missing
keywords should be minimum 5
Figure visibility can be improved
Check grammatical errors
Include latest references
Author Response
We would like to thank the reviewer for their time in reading our manuscript. We have done our best to respond to the nominal comments. The original comments are in italics, and our responses are in bold.
Equations numbering missing
Done
keywords should be minimum 5
Done
Figure visibility can be improved
We are not certain what is meant by figure visibility.
Check grammatical errors
Done
Include latest references
It is unclear what the reviewer is requesting here. Our most recent citations are 2022.
Reviewer 3 Report
This paper develops an assessment technique and subsequent metrics 1 that help to build understanding of the balance between network size and task performance in simple model networks. High aIQ dense and convolutional models were shown to have desireable properties, such as resistance to overtraining and comparable general performance 381 on EMNIST to much larger networks. There are some comments listed as follows:
1 Only LetNet is considered in the experiments. Some large-scale neural networks should be investigated.
2 Only MNIST dataset is considered in the experiments. It seems that the experiments are too simple.
3 There are some errors in the whole text. Please check it.
Some descriptions are not appropriate. For example,
in Line 114, this sentence should not be indented.
Author Response
We would like to thank the reviewer for reading the manuscript and providing feedback. We respond to specific points below, indicating original comments in italics, and responses in bold.
1 Only LetNet is considered in the experiments. Some large-scale neural networks should be investigated.
This is definitely one of the deficits of the current study, as pointed out in the discussion. Also as discussed, there can be some scaling problems to this approach. We are currently working on methods to scale the capturing of state space so that evaluation on larger networks can be performed.
2 Only MNIST dataset is considered in the experiments. It seems that the experiments are too simple.
We agree the experiments are simple. It was unclear when we began this work whether this approach would give a useful metric. Thus, the main novelty of the paper are the introduction of state space and the ability to use it to develop useful metrics. As discussed in the previous comment, we are currently working on better implementations to evaluate these methods on larger datasets and larger models.
3 There are some errors in the whole text. Please check it.
Done.
Round 2
Reviewer 1 Report
1) introduction: "For example, some networks are too large for the task they are being trained to perform, resulting in poor task performance relative to a smaller network trained to perform the same task." These sentence does not sound. Why I should prefer a larger network respect to a smaller one if the smaller network has greater performance?
Authors: The next sentence after what the reviewer quotes in our paper answers the question the reviewer is asking. Large networks that do not have skip connections tend to perform worse that small networks (see reference 5). The reason to prefer large networks is that they tend to have more inference power, but if they are too deep and without skip connections, the performance can be bad.
Reviewer: this answer still let my question open. Again, why anyone should prefer a large trained model if the performance are worse respect to smaller one? It is not important they "tend" to more inference power, if the final trained model perform bad.
3) section 2.1: "When one image is passed through a convolutional neural network, convolutional layers will generate multiple states per image. In contrast, dense layers will generate only one state per image." I don't agree with this point. A convolutional layer can be viewed as a dense layer with shared weights and broken connection, therefore both dense and convolutional layers should, in my opinion, generate the same number of states (in my opinion, just one) per input.
Authors: If we understand the reviewer correctly, the reviewer is correct that each location in an image convolved with a filter has one input per region. However, for an image there are multiple locations that the filter is evaluated. For example, if an image is of size 7x7 and convolved with at 3x3 filter, if no padding is added to the original image then the output of the operation is a 5x5 image. This means there are 25 different states, one for each location, for a single input image. In contrast, a dense layer aggregates all input locations and produces one state.
Reviewer: I don't agree. Please refer, for example, to https://cs231n.github.io/convolutional-networks/#conv to understand why I consider a convolutional network as a fully connected one with some contraints (i.e. shared weights and partially connections), therefore, I expect just one state for each convolutional filter applied to any input.
6) the authors define the the artificial intelligence quotient as the geometric mean (again) of the entropy with the performance (that, I presume, it is assumed between 0 and 1). I think that adopting the term "artificial intelligence quotient" to define this measurement could be misleading, especially if compared with the classical definition of intelligence quotient applied usually, that it is not said that it is a direct consequence of the neural efficiency, since it is usually measured regardless the neural efficiency, considering other factors (such as age, intelligence tests,...). Therefore, defining the proposed measurement as "artificial intelligence quotient" could not be appropriate.
Authors: Thank you for the comment. This is something we wrestled with as well, and the reason we decided to stay with aIQ is that the metric approximates features of human intelligence as described in the introduction. In a previous version we had additional background and discussion about the nuance of what aIQ is really is a reflection of, and that should be more along the lines of crystalized intelligence or task specific intelligence. However, we found that discussion to be fairly meandering and distracting. We maintain artificial intelligence quotient is still an appropriate name, because high values indicate a balance of task performance and efficiency, key features of intelligence in the human brain.
Reviewer: I still don't agree with the answer provided by the authors. efficiency is not said to be the cause of intelligence, maybe it can be a consequence. Furthermore, classical IQ in human brain is computed without referencing to neural "efficiency", therefore, as far as I understand and find reasonable the work done and the proposed measure, at the same time I find inappropriate to define a measurement based on efficiency of artificial NN as aIQ.
Author Response
1) introduction: "For example, some networks are too large for the task they are being trained to perform, resulting in poor task performance relative to a smaller network trained to perform the same task." These sentence does not sound. Why I should prefer a larger network respect to a smaller one if the smaller network has greater performance?
Authors: The next sentence after what the reviewer quotes in our paper answers the question the reviewer is asking. Large networks that do not have skip connections tend to perform worse than small networks (see reference 5). The reason to prefer large networks is that they tend to have more inference power, but if they are too deep and without skip connections, the performance can be bad.
Reviewer: this answer still let my question open. Again, why anyone should prefer a large trained model if the performance are worse respect to smaller one? It is not important they "tend" to more inference power, if the final trained model perform bad.
Authors: The Authors apologize for not providing a better explanation to the reviewers question originally. Network size and what is "large" is relative to the task the network is being asked to perform and the distribution of the data. What the Authors were trying to discuss in this section is that depending on the complexity/difficulty of a task it has been shown that there are "optimal" network sizes to solve a given task. Smaller networks than this "optimal" sacrifice accuracy and inferential certainty, while larger networks than this "optimal" can begin to memorize training data inputs, suffer from modal collapse, and thus perform worse on test data. Therefore, to directly answer the reviewers question, a researcher should/would desire to increase network size (assuming any compute budget is amenable to the task) to provide generalizability, stronger inference certainty, and thus better accuracy (or whatever desired outcome measure) but not to the point that the network has the capacity to memorize training data and/or suffer from modal collapse. Architectural solutions in the form of skip connections, drop out, and normalization layers, etc. as discussed in the introduction, have all been discovered to help researchers increase network size before reaching this "optimal" size, but even given these techniques there are limits on the maximum network size for a given task and dataset. This manuscript seeks to develop a new tool to measure where this "optimal" exists for a given dataset and task. To better convey the above the quoted section has been revised to have the following text:
For example, some networks are too large for the task they are being trained to perform, resulting in poor task performance relative to a smaller network trained to perform the same task. This phenomenon is observed by increased performance on a task as the network depth increases to a point after which performance decreases. One solution to the costly search for the “optimal” network depth is skip or residual connections, which may require depth rescaling using another neuron layer.
3) section 2.1: "When one image is passed through a convolutional neural network, convolutional layers will generate multiple states per image. In contrast, dense layers will generate only one state per image." I don't agree with this point. A convolutional layer can be viewed as a dense layer with shared weights and broken connection, therefore both dense and convolutional layers should, in my opinion, generate the same number of states (in my opinion, just one) per input.
Authors: If we understand the reviewer correctly, the reviewer is correct that each location in an image convolved with a filter has one input per region. However, for an image there are multiple locations that the filter is evaluated. For example, if an image is of size 7x7 and convolved with at 3x3 filter, if no padding is added to the original image then the output of the operation is a 5x5 image. This means there are 25 different states, one for each location, for a single input image. In contrast, a dense layer aggregates all input locations and produces one state.
Reviewer: I don't agree. Please refer, for example, to https://cs231n.github.io/convolutional-networks/#conv to understand why I consider a convolutional network as a fully connected one with some contraints (i.e. shared weights and partially connections), therefore, I expect just one state for each convolutional filter applied to any input.
Authors: We apologize for not making a more clear delineation between the "state" of a macro output and the "state" of a micro output. In re-reading our statement, and your concern, it is ambiguous whether we are referring to a macro state (aka activation map output from a convolution operation) or the micro state of a layer in a network. Your statement is correct when referring to the macro state of a network layer given an input image or image tile, since only one unique set of values is generated for the given input. However, each location in the output of a convolutional layer also contains the specific activation patterns of the neurons in the layer (a “micro” state). We have reworded our statement to better reflect the macro/micro neuronal context we sought to convey in our statement. The below text was edited:
When one image is passed through a convolutional neural network, convolutional layers will generate multiple neuron layer states per image (micro states) that make up a spatial activation map (a macro state). In contrast, dense layers will generate only one neuron layer state per image (the micro and macro state are the same).
6) the authors define the term artificial intelligence quotient as the geometric mean (again) of the entropy with the performance (that, I presume, it is assumed between 0 and 1). I think that adopting the term "artificial intelligence quotient" to define this measurement could be misleading, especially if compared with the classical definition of intelligence quotient applied usually, that it is not said that it is a direct consequence of the neural efficiency, since it is usually measured regardless the neural efficiency, considering other factors (such as age, intelligence tests,...). Therefore, defining the proposed measurement as "artificial intelligence quotient" could not be appropriate.
Authors: Thank you for the comment. This is something we wrestled with as well, and the reason we decided to stay with aIQ is that the metric approximates features of human intelligence as described in the introduction. In a previous version we had additional background and discussion about the nuance of what aIQ is really is a reflection of, and that should be more along the lines of crystalized intelligence or task specific intelligence. However, we found that discussion to be fairly meandering and distracting. We maintain artificial intelligence quotient is still an appropriate name, because high values indicate a balance of task performance and efficiency, key features of intelligence in the human brain.
Reviewer: I still don't agree with the answer provided by the authors. efficiency is not said to be the cause of intelligence, maybe it can be a consequence. Furthermore, classical IQ in human brain is computed without referencing to neural "efficiency", therefore, as far as I understand and find reasonable the work done and the proposed measure, at the same time I find inappropriate to define a measurement based on efficiency of artificial NN as aIQ.
Authors: The Authors thank the Reviewer for their steadfast standpoint and explanation. The authors note that there is no definition, consensus, or approach that is universally accepted to explain human intelligence quotient (IQ). There is no formula or well defined explanation/theorem at any cellular, tissue, physiological, or behavioral level for what IQ is. Given this lack of widely accepted formula or widely understood mathematical definition, creating a distinct acronym, aIQ, and stating its mathematical definition was seen as unambiguous. We agree if there were another term such as respiratory rate (RR), that had a well defined and understood mathematical definition, (breaths/unit time), and we chose to name our factor artificial respiratory rate (aRR) and give it our current definition it could sow confusion in the field. However, to our knowledge there is no such understanding, consensus, or formula that would confuse readers as to what our factor describes and how it is defined. In fact, the Authors believe the current ambiguity of the field about the definition of IQ will lead readers to ask the question "What do the Authors even mean by aIQ?" Driving them to read the methods section carefully and think critically about what we propose instead of glossing over those sections. The Authors see this as a factor in favor of the name rather than a detriment. Lastly, the Reviewer seems to draw a direct relationship between aIQ and human IQ that the Authors do not make or imply in the paper. The Authors found our approach in neurobiological concepts BUT importantly we do not try to draw direct relationships to the assessment of human IQ and its relationship to neural efficiency. Given the above points, the Authors do not think there is a significant justification to change the factor's name in this publication.